# A Diet for Healthy Weight: Why Reaching a Consensus Seems Difficult

**DOI:** 10.3390/nu12102997

**Published:** 2020-09-30

**Authors:** Moul Dey, Purna C. Kashyap

**Affiliations:** 1Department of Health and Nutritional Sciences, Box 2275A South Dakota State University, Brookings, SD 57007, USA; 2Department of Gastroenterology and Hepatology and Department of Physiology and Biomedical Engineering, Mayo Clinic, Rochester, MN 55905, USA; kashyap.purna@mayo.edu

**Keywords:** dietary guidelines, nutrition research, obesity

## Abstract

Overweight and obesity are global health problems that contribute to the rising prevalence of non-communicable diseases, such as type 2 diabetes, heart disease, and certain cancers. The World Health Organization recognizes obesity as a primarily diet-induced, preventable condition, yet losing weight or keeping weight loss permanent is a universal challenge. In the U.S., formal dietary guidelines have existed since 1980. Over the same time-period, the incidence of obesity has skyrocketed. Here, we present our perspective on why current dietary guidelines are not always supported by a robust body of scientific data and emphasize the critical need for accelerated nutrition research funding. A clear understanding of the interaction of dietary patterns with system-level biological changes in a precise, response-specific manner can help inform evidence-based nutrition education, policy, and practice.

## 1. The Unmet Goal

Nutritional health research takes the science of nutrition a step further—beyond the goal of providing the human body with necessary energy and nourishment—to promote health by treating or reducing risk of diseases. Advances in nutritional health knowledge in the past century helped reduce vitamin-deficiency diseases, such as beriberi (vitamin B1), pernicious anemia (vitamin B12), scurvy (vitamin C), and rickets (vitamin D) [1]. However, this early success of nutritional health research did not translate into curbing the burden of complex non-communicable diseases. For example, obesity is a common denominator in many of these diseases. Despite widespread weight-management guidelines, diets, and programs, 2 in 3 adults and 1 in 6 children are obese or overweight in the U.S. [2,3,4,5]. As the prevalence of obesity and related chronic diseases continue to increase, a one-size-fits-all solution seems unlikely.

The discussion in this article is primarily focused on the specific context of the U.S. The chronic health crisis has received substantial attention from the government and national agencies over a long period of time. Differential metabolic contributions of energy-yielding macronutrients were recognized early on. However, the debate continues to-date regarding an optimal macronutrient ratio for weight-control [6,7,8,9,10,11]. Back in 1977, a Senate committee report on “Dietary Goals for the United States” recommended a low-fat, low-cholesterol diet for Americans [12]. This was the basis for future dietary guidelines for Americans (DGA) released since 1980 by the Department of Agriculture and the Department of Health and Human Services. In addition, the Healthy People Program was developed in 2001 by the Food and Drug Administration and the National Institutes of Health to specifically address the rising trend in metabolic diseases associated with diet and weight gain [13]. Despite these and other nation-wide efforts, the goal to harness the power of nutrition for obesity mitigation (Figure 1) and chronic health promotion is at best partially fulfilled [2,3,4,5].

## 2. The Problem

In 2017, a congressionally mandated report from the National Academies of Sciences, Engineering, and Medicine called for a comprehensive redesign of the process for updating the future editions of DGA. The DGA revision process begins with an independent evaluation of the scientific evidence by the Dietary Guidelines Advisory Committee every five years. However, given the DGA was not always informed by a robust body of research data, strengthening of scientific rigor was identified as one of the needs in the report [14]. For example, not until recently, investigators from the U.S. Department of Agriculture have put their recommended diet through the rigorous scientific test i.e., controlled-feeding, double-blind, randomized intervention to evaluate its impact on cardiometabolic disease risks. Contrary to the generally held perception, the DGA-based test-diet did not improve glucose homeostasis or lipid profiles in the at-risk study cohort [15]. Limitations of the research data used to develop nutrition policies and guidelines were also being discussed [14,16,17]. The quality of commonly utilized observational nutrition research designs and those involving dietary recall have been questioned over the years [6,7,8,9,10,11,18]. Such controversies can potentially erode public trust in nutrition advice. However, it is important to acknowledge that designing robust nutrition studies can be challenging for many reasons. Below, we discuss the ones we believe are the most important.

Traditional studies frequently collected observational and correlational data. Even when dietary interventions were considered to indicate causality, simply borrowing from double-blind randomized controlled drug-trial designs may be less than adequate. Diet constitutes an extremely complex pattern of exposures in comparison to single entity small molecules. Thus, many social, behavioral, and environmental factors that would not otherwise confound drug-response can affect response to diets. Moreover, defining an appropriate control group or the blinding protocol for a diet-trial is less straight forward than in the case of a pharmaceutical trial. In addition, response to food is more subtle and slow compared to the response to drugs. This makes it all the more essential to assess the molecular mechanisms underlying an observed metabolic phenotype to help distinguish a random correlation from a true diet-induced clinical outcome. In post-genomic years, practical molecular tools (e.g., transcriptomic, metabolomic, epigenomic, and metagenomic measurements, etc.) are becoming more routine to capture the biological complexities underlying clinical observations. A wealth of new information on factors influencing physiological responses to diet such as the gut microbiome has introduced a new frontier in nutrition research that may hold the potential to clarify at least some unknowns. Early studies, including our own, have presented the possibility that microbiome profiling and multi-omic data integration could assist clinical evaluation and help predict responses to diet-therapies for individuals [19,20,21,22,23].

Diet trials call for randomized controlled metabolic feeding studies with built-in experimental measures to mitigate some of the additional challenges faced by nutrition research as discussed above. A larger body of independent research that incorporates longitudinal sampling as well as different populations in terms of age, lifestyle, health-status, and genetic background would also help elevate the quality of generated data. However, randomized controlled feeding studies with mechanistic, behavioral, and clinical endpoints—considered the gold standard in diet research—are less frequently performed. They are expensive and logistically challenging to administer and can suffer from lack of inter-disciplinary expertise of the investigating team or more critically, low funding. Despite the tremendous potential of scientific knowledge generated by well-designed studies to benefit public health combined with the greater complexity and heterogeneity that nutrition research presents, public funding for nutrition science has been consistently low. For example, only 5% of the National Institutes of Health’s annual budgets went toward nutrition research between 2012–2016 [24]. The share of private industry funding for research on whole diets and dietary patterns is also significantly smaller than for drug or supplement discovery, likely due to potentially less profitable intellectual properties. Of note, the role of industry funding toward generating credible research data is a topic by itself for a different time [25]. However, another point worth noting here is that most funding agencies prize paradigm-shift over incremental research. While innovation is critical, incremental nutrition studies can potentially allow convergence (or lack thereof) of data to build a strong scientific knowledge-base for informing effective public health policies. 

## 3. Lessons Learned

While diet is not the only factor to be considered in the context of obesity, it is a critical one. Thus, looking at the prevalence data, one can argue that the past and current versions of the DGA have been less than adequate for controlling obesity (Figure 1). However, there may be lessons to learn from this apparent lack of success. A major concern has been the premature reliance on diet-related health outcome data from smaller or poorly designed studies. For example, early ecological studies that did not establish causality, nevertheless, contributed to the widespread belief that dietary fats and cholesterols are major contributors to heart diseases [18,26,27]. In 1980, the Food and Nutrition Board of the National Academy of Sciences noted that the predominantly correlative and observational scientific evidence was insufficient to support a generic low fat, low cholesterol lifestyle for all Americans [28]. Yet, four decades later, heart disease is the leading cause of mortality according to the Centers for Disease Control and Prevention while we continue the emphasis on low-fat dietary patterns and our discussions on the “lack of strong evidence” issue in nutrition research knowledge. 

The obesity epidemic associates with the rising prevalence of type 2 diabetes (T2D) [29]. T2D is generally perceived as a progressive disease, but growing evidence may challenge the status quo in the near future. In 2017, a trial reported that T2D remission is possible through dietary weight management administered systematically by a primary health-care team [30]. Since then, additional studies showed that primary care-led, physician-involved, dietary and lifestyle interventions can effectively treat and reverse type 2 diabetes and associated comorbidities [30,31,32,33,34,35]. Many of these studies used a low-carbohydrate, high-fat diet that is currently neither approved nor consistently advocated by the Department of Agriculture, the Food and Drug Administration, the American Heart Association, the American Diabetes Association, or the Academy of Nutrition and Dietetics. Alongside clinical evidence, some historical, epidemiological, and early mechanistic data indicate that high carbohydrate intake and generic fat restriction, as encouraged by the DGA may not promote long term health as was originally believed [8,10,36]. Changing a long-standing position on macronutrient recommendations for the general public will require additional research and deliberations. However, the body of evidence presented above may support immediate, medically supervised, implementation of low-carbohydrate diet-therapy for obesity and T2D formulated on a case-by-case basis depending on individual patient needs.

## 4. Rethinking Strategies 

Patients and consumers are more interested than ever in lifestyle medicine with nutrition advice at its core; however, such a healthcare model is not routine at present [37]. Remission in patients and prevention in at-risk population through lifestyle interventions can reduce obesity-related healthcare expenditures estimated to be over 147 billion dollars annually [38]. To switch to a nutritionally directed healthcare model, accelerated nutrition research leading to deeper scientific understanding of the biological drive for weight gain will be critical as will the availability of trained physicians willing to adopt and implement nutritional interventions. 

Physicians are trained to treat complications arising from obesity, such as heart disease and T2D, but medical school curriculum emphasizes much less on the utilization of nutritional interventions to mitigate the risks of developing those complications in the first place [39,40]. As discussed earlier in the article, only recently has there been compelling evidence that primary care physician-led dietary interventions can effectively treat and reverse metabolic diseases [30,31,32,33,34,35]. More commonly, unless drugs or surgical interventions are considered, patients are referred to retail programs run by registered dieticians and health coaches to deal with body weight problems. 

Dieticians and fitness coaches are experts in the implementation process of the national guidelines within the social context of community needs and cultures. However, the current training and education protocols do not provide these professionals the understanding of the biological basis of weight gain, loss, and regain, and most never question why the DGA does not work for some clients. In a recent study, we reported a large variation in weight-loss response within a retail weight-management program that use the current guidelines. We observed a correlation between the intestinal microbiota and differential response. Because the health coaches and dieticians involved with retail programs are not trained or required to keep up with newer research, they are often unable to set realistic goals based on client features such as sex, baseline body composition, or microbiota profile [21]. Personalizing intervention strategies to maximize response has the potential to improve adherence and can contribute to improved effectiveness of any program. 

Adherence to diet-therapies and lifestyle interventions may be influenced by complex factors such as overall socio-economic circumstances, insurance coverage for weight-loss programs, food preference, neurobiology of food choice, and beyond. Efforts to assess these less-understood behavioral and environmental factors in clinical research as well as within community settings are necessary. However, we would argue that two factors that are better understood and can help improve adherence are setting realistic goals and ensuring progress early on. For example, research shows that less than expected weight-loss during the initial weeks may lower the motivation to adhere and can lead to large dropouts. The studies also report cost, scheduling conflicts, tiring of the limited food choices, unrelated health issues, or lack of consistent/sustained results as other potential reasons for discontinuing weight-loss interventions [41,42,43].

## 5. Looking Forward

Scientific knowledge must build on converging data from interdisciplinary studies that are carried out independently in a range of different environmental settings and populations. Proof-of-concept observational human data or mechanistic model animal findings must be reconfirmed by randomized controlled metabolic feeding trials that incorporate phenome-wide strategies and intermediate pathway endpoints. The optimal macronutrient composition of a diet-therapy will depend on factors such as appetite, thermogenesis, and energy homeostasis as well as on the interaction of these factors with an individual’s gut microbiota. Well-designed studies can help push the field towards recognition of the complex biological effects of foods and dietary patterns, emphasizing quality, quantity, and mutual interactions among nutrients. It is increasingly recognized that macronutrient quality (e.g., fiber versus simple sugars within the carbohydrate group) and non-nutritive phytochemicals as well as food structure, preparation, and processing are all important factors to consider. Therefore, in addition to metabolic and hormonal regulations, food preferences, clinical history, and other lifestyle patterns and behaviors may be considered. 

A reevaluation of nutrition research funding to enable high-quality scientific data as well as the integration of a nutrition culture in mainstream healthcare practice holds the promise to reduce the financial, societal, and personal burden of obesity and diet-related chronic diseases. Collective actions from public health stakeholders at every level would be key to let the healing power of food unfold like never before.

## Figures and Tables

**Figure 1 nutrients-12-02997-f001:**
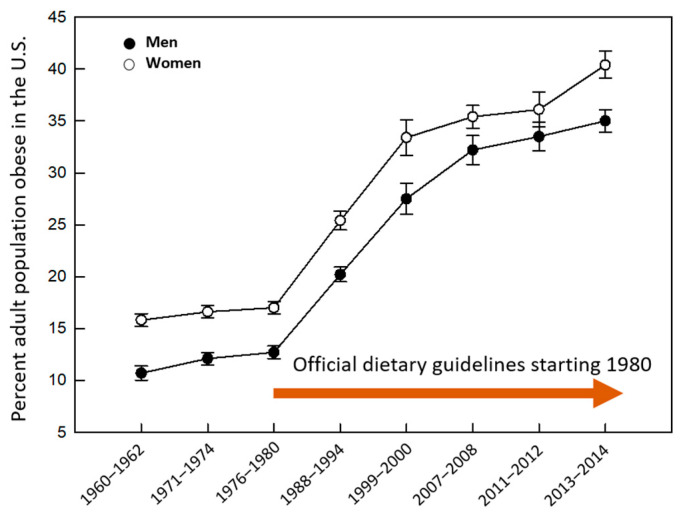
Age-adjusted prevalence of obesity by sex in the United States, 1960–2014. Data for ages 20–74. Overweight population and pregnant females were excluded. Graph generated using publicly available data from the National Center for Health Statistics, the National Health Examination Survey, and the National Health and Nutrition Examination Survey.

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
