# Peer review of "A Diet for Healthy Weight: Why Reaching a Consensus Seems Difficult"

_nutrients, 2020, doi:10.3390/nu12102997_

Round 1
Reviewer 1 Report
The premise of this piece is that our national dietary guidelines are not supported by research. This is an interesting contribution to the conversation. While I do not know that everyone in the academic community would agree with everything in this piece, it is well-written overall and brings an interesting perspective. The commentary could benefit from considering the following revisions related to reducing stigma and clarifying some terms/points:
On line 23 (first sentence of the commentary), there is no reference. The statement:
"Nutritional health research during the past century helped reduce or in some cases eliminate calorie malnutrition and micronutrient-deficiency diseases, especially in the high-income nations” seems far too strong, particularly the use of the word “eliminate”.
Relatedly, it might be good to define “nutritional health research” and to specify which “malnutrition and micronutrient-deficiency diseases” the author is referring to.
It is suggested that the authors use person first language when talking about obesity throughout (e.g., “continue to have obesity” and not “continue to be obese”)
The cartoons in figure 1 should be eliminated. They are stigmatizing and shaming without adding anything to the data.
Author Response
Response to reviewer #1 comments:
The premise of this piece is that our national dietary guidelines are not supported by research. This is an interesting contribution to the conversation. While I do not know that everyone in the academic community would agree with everything in this piece, it is well-written overall and brings an interesting perspective. The commentary could benefit from considering the following revisions related to reducing stigma and clarifying some terms/points:
- On line 23 (first sentence of the commentary), there is no reference. The statement: "Nutritional health research during the past century helped reduce or in some cases eliminate calorie malnutrition and micronutrient-deficiency diseases, especially in the high-income nations” seems far too strong, particularly the use of the word “eliminate”. Relatedly, it might be good to define “nutritional health research” and to specify which “malnutrition and micronutrient-deficiency diseases” the author is referring to.
We have addressed all the points raised by the reviewer. 1) toned down the statement that appeared strong. 2) added reference. 3) clarified what nutritional health research entails. 4) cited many examples of micronutrient deficiency diseases. Please see lines 23 through 28 for the said changes.
- It is suggested that the authors use person first language when talking about obesity throughout (e.g., “continue to have obesity” and not “continue to be obese”)
We have made this type of language change throughout the text.
- The cartoons in figure 1 should be eliminated. They are stigmatizing and shaming without adding anything to the data.
We have removed the cartoons. Thank you.
Reviewer 2 Report
Thank you very much for the opportunity to review this manuscript addressing the low efficacy of current dietary guidelines in obesity mitigation. The manuscript is well written and it helps in understanding why one-size-fit-all solution seems unlikely in the face of rising prevalence of obesity. The authors also point out relevant directions for future research.
I have some comments for the authors to address:
TITLE
The title is appropriate
ABSTRACT
The abstract is very well and include the most relevant information of the work.
- UNMET GOAL
In this section the authors call attention for the rising prevalence of obesity in U.S. in spite of widespread weight-management guidelines. In my view, the authors should clarify whether this conclusion is limited to U.S. or is worldwide. If this is a global problem, data from other countries/continents should be added as example.
- THE PROBLEM
Page 2; line 53: When the authors state that “the quality of commonly utilized nutrition research designs has been questioned” it would be pertinent to include more information about the nature of the limitations of the studies.
Page 2; lines 62-63: In my view, psychological factors (e.g., emotion regulation, affect, motivation) should also be taken into account.
Page 2; lines 62-66: The authors stated that the complexity of eating behavior limited the conclusions from RCTs and suggested the implementation of “RCT metabolic feeding studies with built-in experimental measures” to mitigate this challenge. It is not clear how this suggestion would allow to address social, behavioral, and environmental factors linked to eating behavior. Could the authors further develop this argument?
- LESSONS LEARNED
Page 3; lines 91: Why is the scientific evidence insufficient to support a generic low-fat diet? What are the studies flaws?
Page 3; lines 92-93: Can the high prevalence of heart disease disprove the efficacy of low-fat policy? Whether or not the individuals follow this guideline also plays into the equation. Neurobiology of food choice (which is different from food preference) is an important field of research that needs to be addressed to understand obesity.
Page 3; lines 108-111: In my view, the meaning of the last part of the sentence is not absolutely: “…the immediate take away may be the potential of medically implemented diet-therapy for reversal of obesity and T2D.”
- RETHINKING STRATEGIES
Page 4; line 135; Could the authors exemplify some “clinical features” they are thinking about?
- LOOKING FORWARD
Assuming that the optimal macronutrient composition of a diet depends on a variety of factors, do the authors consider that dietary guidelines to control obesity should be banned? The authors’ position regarding this issue would enrich the manuscript.
Author Response
Response to reviewer #2 comments
Thank you very much for the opportunity to review this manuscript addressing the low efficacy of current dietary guidelines in obesity mitigation. The manuscript is well written and it helps in understanding why one-size-fit-all solution seems unlikely in the face of rising prevalence of obesity. The authors also point out relevant directions for future research.
I have some comments for the authors to address:
TITLE: The title is appropriate
ABSTRACT: The abstract is very well and include the most relevant information of the work.
- Thank you for your kind comments.
- UNMET GOAL
In this section the authors call attention for the rising prevalence of obesity in U.S. in spite of widespread weight-management guidelines. In my view, the authors should clarify whether this conclusion is limited to U.S. or is worldwide. If this is a global problem, data from other countries/continents should be added as example.
- We have clarified this information in line #34.
- THE PROBLEM
Page 2; line 53: When the authors state that “the quality of commonly utilized nutrition research designs has been questioned” it would be pertinent to include more information about the nature of the limitations of the studies.
- We tried to clarify this specifically in this paragraph as the reviewer suggested. Please see line #63-64, plus some other pertinent changes throughout this paragraph. However, we had dedicated (original version) an entire paragraph following the paragraph in question describing the nature of the limitations. Please see this next paragraph starting line #68.
Page 2; lines 62-63: In my view, psychological factors (e.g., emotion regulation, affect, motivation) should also be taken into account.
- In line #71-73, we mentioned (original version) social and behavioral aspects are especially important for nutrition research and should be taken into account. Furthermore, we have now added a small paragraph starting at line #163 (later in the article) discussing in greater details why social and behavioral factors matter in diet research in the context of adherence.
Page 2; lines 62-66: The authors stated that the complexity of eating behavior limited the conclusions from RCTs and suggested the implementation of “RCT metabolic feeding studies with built-in experimental measures” to mitigate this challenge. It is not clear how this suggestion would allow to address social, behavioral, and environmental factors linked to eating behavior. Could the authors further develop this argument?
- We have addressed this comment by taking multiple measures. 1) We have moved this statement in question “…. with built-in experimental measures” to the next paragraph that now starts at line#86. There we added few subsequent lines to clarify the reviewer’s question (line #88-91). 2) Please also see the new information added in the same context starting at line #163.
- LESSONS LEARNED
Page 3; lines 91: Why is the scientific evidence insufficient to support a generic low-fat diet? What are the studies flaws?
- We believe we had a sentence prior to the statement (in original version) that answers this question. Please see line #109-111 that states: “…early ecological studies did not establish causality, nevertheless, contributed to the widespread belief that dietary fats and cholesterols are major contributors to heart diseases 18, 25, 26.”
- Further we have added few words in the sentence in question to make it even more clear. Please see line #112-113 that now states: “…. the predominantly correlative and observational scientific evidence was insufficient to support a generic low fat, low cholesterol lifestyle for all Americans 27.”
Page 3; lines 92-93: Can the high prevalence of heart disease disprove the efficacy of low-fat policy? Whether or not the individuals follow this guideline also plays into the equation. Neurobiology of food choice (which is different from food preference) is an important field of research that needs to be addressed to understand obesity.
- The reviewer’s comment reminded us to add a brief discussion on adherence starting at line #163. Neurobiology of food choice relates to adherence, which is also mentioned in this new paragraph on adherence.
Page 3; lines 108-111: In my view, the meaning of the last part of the sentence is not absolutely: “…the immediate take away may be the potential of medically implemented diet-therapy for reversal of obesity and T2D.”
- This sentence is reworded for greater clarity, please see line #131-134. Some wording changes are also made to the preceding sentence to further bring out the point made in the sentence in question.
- RETHINKING STRATEGIES
Page 4; line 135; Could the authors exemplify some “clinical features” they are thinking about?
- This statement relates to our recently published work. We have added examples of the features in line # 154-158: “…features such as sex, baseline body composition, or microbiota profile 21.”
- LOOKING FORWARD
Assuming that the optimal macronutrient composition of a diet depends on a variety of factors, do the authors consider that dietary guidelines to control obesity should be banned? The authors’ position regarding this issue would enrich the manuscript.
- We believe without greater amount of better quality nutrition research data becoming available related to DGA, it will be difficult to answer this question in a precise manner. At present, the guidelines for the general public must be revised to reflect most recent research efforts and directions such at the data on the gut microbiota-diet interactions. Two important points are that A) we must have a greater nutrition culture in mainstream healthcare and the 2) recognition that the guideline does not work for all. These points are tied to the evidence-based suggestion that in parallel to generic guidelines, physician-supervised implementation of diet-regimens can and should differ from the guidelines in a patient’s need-specific manner. [Physician-supervision is important for both safety and efficacy monitoring especially when the diet-therapy is deviating from general guidelines for obese individuals with serious heart disease or diabetes.] We discuss these points throughout the article, but more specifically in the following lines: #52-56, #101-104, #129-134, #174-191.